# TCR-EML: Explainable Model Layers for TCR-pMHC Prediction

## Abstract

T cell receptor (TCR) recognition of peptide-MHC (pMHC) complexes is a central component of adaptive immunity, with implications for vaccine design, cancer immunotherapy, and autoimmune disease. While recent advances in machine learning have improved prediction of TCR-pMHC binding, the most effective approaches are black-box transformer models that cannot provide a rationale for predictions. Post-hoc explanation methods can provide insight with respect to the input but do not explicitly model biochemical mechanisms (e.g. known binding regions), as in TCR-pMHC binding. "Explain-by-design" models (i.e., with architectural components that can be examined directly after training) have been explored in other domains, but have not been used for TCR-pMHC binding. We propose explainable model layers (TCR-EML) that can be incorporated into protein-language model backbones for TCR-pMHC modeling. Our approach uses prototype layers for amino acid residue contacts drawn from known TCR-pMHC binding mechanisms, enabling high-quality explanations for predicted TCR-pMHC binding. Experiments of our proposed method on large-scale datasets demonstrate competitive predictive accuracy and generalization, and evaluation on the TCR-XAI benchmark demonstrates improved explainability compared with existing approaches.

## 1 Introduction

For the adaptive immune system, T cells are essential for detecting and responding to antigens from pathogens such as viruses, bacteria, and cancer cells (Joglekar & Li, 2021), as well as in autoimmune contexts. The final step of T cell activation involves binding between a peptide presented by the Major Histocompatibility Complex (pMHC) and the T cell receptor (TCR). The specificity of this interaction is the foundation of T cell-mediated immunity and is a major focus of research in both therapeutic development and the study of immune mechanisms. A detailed understanding of T cell response is critical for designing vaccines that provide durable immunity and for developing effective personalized cancer treatments (Rojas et al., 2023; Poorebrahim et al., 2021).

CD8+ T cells are activated through the MHCI pathway, whereas CD4+ T cells are activated through the MHCII pathway. Epitope prediction for CD8+ T cells has achieved notable success, while mechanisms of CD4+ T cell response remain less well understood. The CD4+ T cell response can be viewed as a two-stage recognition process. In the first stage, antigens are processed by antigen-presenting cells (APCs) and loaded onto MHCII molecules, which are subsequently presented on the APC surface (Davis & Bjorkman, 1988; Neefjes et al., 2011). In the second stage, TCRs recognize these pMHC complexes, initiating the T cell response. TCR recognition is mediated by the $\alpha$ and $\beta$ domains, each composed of variable (V), joining (J), and constant (C) regions, with the $\beta$ chain also containing a diversity (D) region (Bosselut, 2019). Accurate prediction of T cell responses therefore requires modeling both antigen processing and TCR-pMHC binding (Peters et al., 2020; Nielsen et al., 2020).

Early computational work in epitope prediction emphasized peptide-MHCII binding using allele-specific machine learning methods (Nielsen et al., 2020), with tools including NetMHCpan (Hoof et al., 2009; Nielsen et al., 2007) and NetMHCcons (Karosiene et al., 2012). More recently, antigen processing has been modeled with the Antigen Processing Likelihood (APL) algorithm (Mettu et al.,

2016; Bhattacharya et al., 2023; Li et al., 2024a;b; Charles et al., 2022), which accounts for structural features that determine which peptides are presented by MHCII molecules.

Prediction of TCR-pMHC binding remains a central challenge in quantitative immunology and adaptive immunity (Hudson et al., 2023). Both unsupervised and supervised learning algorithm approaches have been explored (Hudson et al., 2023; 2024). Unsupervised methods cluster TCR sequences using similarity metrics such as TCRdist3 (Mayer-Blackwell et al., 2021) applied to complementarity-determining regions (CDRs), without requiring binding labels or epitope information (e.g., GIANA (Zhang et al., 2021), GLIPH2 (Huang et al., 2020)). The resulting clusters are then used to support experimental analysis (Hudson et al., 2024). Supervised methods leverage large TCR-pMHC datasets (Hudson et al.,

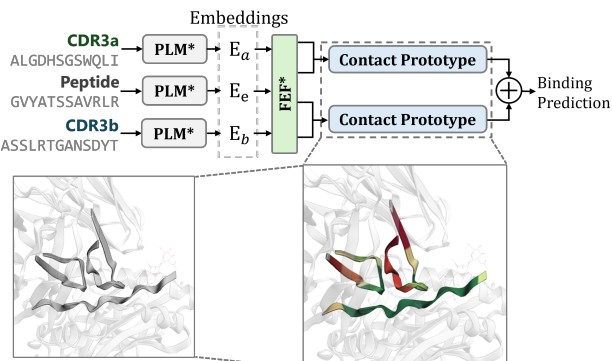

Figure 1: The explainable model layers include a Feature Enhancement and Fusion (FEF) block and contact prototype layers, which not only predict TCR-pMHC binding but also provide contact scores corresponding to contact distances. In the absence of experimental TCR-pMHC structures, the contact prototype illuminates TCR-pMHC binding patterns.

2023) from sources including VDJdb (Bagaev et al., 2020), McPAS-TCR (Tickotsky et al., 2017), and IEDB (Vita et al., 2019), and employ deep learning models such as MixTCRpred (Croce et al., 2024), NetTCR2.2 (Jensen & Nielsen, 2023), TULIP (Meynard-Piganeau et al., 2024), and EGM (Li et al., 2025a).

These models, however, operate as black boxes, and their lack of explainability hinders biological insight into T cell recognition. To address this, post-hoc explanation methods (Kenny et al., 2021) such as QCAI (Li et al., 2025b) and TEPCAM (Chen et al., 2024) have been proposed. These methods demonstrate that deep models can capture mechanisms of TCR-pMHC binding and generate rational predictions (Li et al., 2025b;a). However, post-hoc explanations are not always faithful and have known limitations when applied to black-box models (Rudin, 2019).

To address these challenges we have developed an explain-by-design prediction head for TCR-pMHC modeling that can be used with PLM backbones (e.g., ProteinBERT (Brandes et al., 2022), ESM-1b Rives et al. (2021), and ESM-2 (Lin et al., 2023)). Our approach makes these widely used models interpretable, without retraining the entire architecture. Our design for TCR-pMHC binding incorporates contact prototypes that can be interrogated after training to reveal mechanistic insights. We evaluate our approach on large-scale TCR-pMHC binding datasets for predictive accuracy and generalization, and on the TCR-XAI benchmark (Li et al., 2025b) for explainability, where it achieves superior performance to existing models.

## 2 BACKGROUND AND RELATED WORK

In this section, we first provide a formal definition of the TCR-pMHC binding problem. We then give an overview of existing deep learning models for TCR-pMHC prediction, protein language model (PLM) backbones and existing explain-by-design methods in deep learning.

### 2.1 TCR-PMHC BINDING PROBLEM DEFINITION

The TCR-pMHC binding prediction task can be expressed as a binary classification problem: given TCR sequences $\text{CDR}_\alpha$ and $\text{CDR}_\beta$ (e.g., from single-cell sequencing), and a peptide sequence $e$, our goal is to predict whether the peptide-MHC complex binds the TCR sequences to elicit a T cell response.

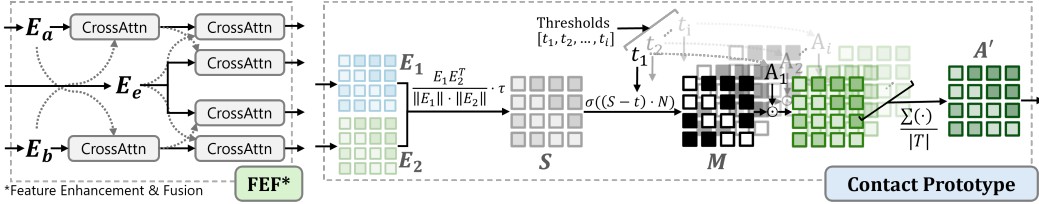

Figure 2: Overview of the our explainable model layers for TCR-pMHC binding prediction. The Feature Enhancement and Fusion (FEF) block integrates information between TCR chains and TCR-peptide pairs. Contact prototype layers model residue-level contact areas and distances between CDR3 regions and the peptide.

## 2.2 SUPERVISED TCR-PMHC PREDICTION MODELS

Transformer models have achieved strong performance across many domains and are increasingly applied to TCR-pMHC binding prediction. In recent work, MixTCRpred (Croce et al., 2024) employs a transformer architecture that incorporates all CDR regions from both TCR $\alpha$ and $\beta$ chains. TULIP (Meynard-Piganeau et al., 2024) is another recent approach that uses an encoder-decoder transformer that is trained using positive data; it has been shown to outperform the widely used NetTCR-2.2 model. Finally, EGM (Li et al., 2025a) is a multi-modal transformer model designed using post-hoc explainability methods that achieves excellent performance. All of these methods are "black-box" and rely on post-hoc explanation methods (Li et al., 2025b) to examine true positive or true negative predictions. PISTE (Feng et al., 2024) incorporates a sliding attention mechanism inspired by binding mechanisms, which provide limited explainability, but requires additional steps to extract explanations. TEIM (Peng et al., 2023) provides residue-level contact scores between TCR and epitope, but relies on structural data for model fine-tuning. A further limitation is that these models are trained from scratch and thus do not use the wealth of data incorporated into pretrained protein language models.

## 2.3 PROTEIN LANGUAGE MODELS

The pretrained foundation models have been applied to various areas and achieved outstanding success, such as in cognitive science (Yin et al., 2025) and image recognition (Radford et al., 2021). To obtain richer representations of protein amino acid sequences, several pretrained, self-supervised, transformer-based foundation protein language models (PLMs) have been developed. ProteinBERT is trained on protein sequences and functional annotations, capturing both local and global features for downstream prediction tasks (Brandes et al., 2022). ESM-1b is a large-scale transformer pretrained on UniRef50, providing contextualized protein embeddings widely used for structure and function prediction (Rives et al., 2021). ESM-2 improves upon this family with larger architectures and expanded pretraining, yielding stronger representations across diverse biological applications (Lin et al., 2023).

## 2.4 EXPLAIN-BY-DESIGN MODELS

Compared to post-hoc approaches, explain-by-design models provide faithful explanations directly through their architecture, without requiring additional operations (Rudin, 2019). These models enable the construction of human-interpretable explanations by integrating explainability into the design of the model itself. Two widely studied approaches are concept bottleneck models (Koh et al., 2020) and prototype learning (Chen et al., 2019). Concept bottleneck models learn a set of human-understandable concepts, derived from properties of the input data, and base their predictions on these concepts (Koh et al., 2020; Yuksekgonul et al., 2022). Prototype learning instead identifies representative prototypes that summarize feature patterns directly from data, and then uses these prototypes to guide decision-making (Chen et al., 2019; Nauta et al., 2023).

## 3 OUR APPROACH

Existing transformer-based TCR-pMHC prediction models use linear classification layers to predict whether the input peptide binds to the input TCRs. These approaches do not provide any internal mechanisms to explain the output classification. Our approach uses inherently explainable model layers along with pretrained protein language model components that provide not only explainable predictions but also improved generalization over existing methods.

Our design consists of two components: (1) feature enhancement and fusion, and (2) contact prototype layers. These components can be directly attached to PLM backbones, which provide embeddings for CDR3a, CDR3b, and peptide sequences, denoted as $E_a \in \mathbb{R}^{N \times d}$, $E_b \in \mathbb{R}^{N \times d}$, and $E_e \in \mathbb{R}^{N \times d}$, respectively, where $N$ is the maximum sequence length and $d$ is the embedding dimension.

### 3.1 FEATURE ENHANCEMENT AND FUSION

Li et al. (2025a) proposed a method that utilizes post-hoc analyses to guide transformer model design, emphasizing the role of cross-attention in TCR-pMHC binding prediction. Cross-attention allows the model to capture interactions within TCR chains as well as between TCR and pMHC, thereby improving mechanistic understanding. Since different pre-trained PLMs adopt diverse architectures and are developed for general-purpose protein modeling, there is no guarantee that the embeddings of CDR3a, CDR3b, and peptide are effectively fused. To address this, we introduce a feature enhancement and fusion (FEF) module that integrates multiple cross-attention layers, motivated by the design principles of Explanation-Guided Model (EGM) (Li et al., 2025a).

Formally, we denote cross-attention from $a$ to $b$ as $\mathcal{A}(Q = a, K, V = b)$, where $a$ serves as the query and $b$ as the key and value. Guided by EGM, we first derive cross-fused representations of CDR3a and CDR3b using:

$$E_{a \to b} = \mathcal{A}(Q = E_a, K, V = E_b), \quad E_{b \to a} = \mathcal{A}(Q = E_b, K, V = E_a). \tag{1}$$

Subsequently, the peptide embeddings are fused with $E_{a \to b}$ and $E_{b \to a}$ to obtain enriched features for TCR-pMHC modeling:

$$E_{e \to a \to b} = \mathcal{A}(Q = E_e, K, V = E_{a \to b}), \quad E_{a \to b \to e} = \mathcal{A}(Q = E_{a \to b}, K, V = E_e), \tag{2}$$

$$E_{e \to b \to a} = \mathcal{A}(Q = E_e, K, V = E_{b \to a}), \quad E_{b \to a \to e} = \mathcal{A}(Q = E_{b \to a}, K, V = E_e). \tag{3}$$

### 3.2 CONTACT PROTOTYPE LAYERS

Residue-level contacts between TCR and pMHC are a key determinant of binding specificity. TCRdist, a widely used method for TCR-pMHC prediction, defines similarity as a weighted mismatch distance between potential pMHC-contacting loops of two receptors (Dash et al., 2017). Similarly, PISTE incorporates TCR-pMHC contact rules into the attention mechanism to improve both predictive performance and explainability (Feng et al., 2024). Motivated by these approaches, we design prototype-based layers to explicitly model contacts between TCR and pMHC using the fused features from FEF.

These layers estimate residue contacts between CDR3a and peptide, and between CDR3b and peptide, respectively. For each pair, the fused embeddings are denoted as $E_1 \in \mathbb{R}^{N \times d}$ and $E_2 \in \mathbb{R}^{N \times d}$. The contact prototype layers take them as inputs and calculate the contact area between these chains. Inspired by the cross-attention mechanism, we model contact distance through similarity:

$$S = \frac{E_1 \cdot E_2^\top}{\|E_1\| \cdot \|E_2\|} \cdot \tau \in [0, 1]^{N \times N}, \tag{4}$$

where $\tau \in \mathbb{R}^+$ is a trainable temperature parameter. Higher similarity corresponds to shorter contact distance. We introduce a set of thresholds $T = [t_0, t_1, ..., t_{|T|}]$ to filter potential contacts, where $|T|$ is the number of thresholds and each $t_i \in [0, 1]$. For threshold $t_i$, residues with similarity greater than $t_i$ are considered to be in contact. To ensure differentiability, we approximate this contact filter as:

$$M_i = \sigma((S - t_i) \cdot N) \in [0, 1]^{N \times N}, \tag{5}$$

where $\sigma$ is the sigmoid function. Following the principle that shorter distances generally imply larger contact areas, we define contact areas under each threshold $t_i$ using $A = \text{softmax}(T) \in [0,1]^{|T|}$. The aggregated contact area is then computed as:

$$A' = \frac{1}{|T|} \sum_{i=1}^{|T|} \sqrt{M_i \odot A_i} \in [0,1]^{N \times N}, \tag{6}$$

where $A'_{k,j}$ denotes the contact area between residues $k$ and $j$, and $\odot$ is element-wise product. The overall contact area between the embeddings $E_1$ and $E_2$ is defined as:

$$w_{1,2} = \frac{1}{N^2} \sum_{k=1}^{N} \sum_{j=1}^{N} A'_{k,j} \in [0,1], \tag{7}$$

where $N^2$ is the maximum possible contact area. For notation clarity, we define a contact prototype function $f : \mathbb{R}^{N \times d} \times \mathbb{R}^{N \times d} \to [0,1]$ with $f(E_1, E_2) = w^c_{1,2}$. Finally, the contact areas between CDR3a and peptide, and CDR3b and peptide, are given by:

$$w_{a,e} = f(E_{e \to b \to a}, E_{b \to a \to e}), \quad w_{b,e} = f(E_{e \to a \to b}, E_{a \to b \to e}). \tag{8}$$

The final contact score for a TCR-pMHC pair is summarized as:

$$\hat{y} = \frac{w_{a,e} + w_{b,e}}{2}. \tag{9}$$

Since $w$ serves as a direct indicator of TCR-pMHC binding, it is optimized using a class-weighted cross-entropy loss that accounts for the positive-to-negative ratio in the training data:

$$\mathcal{L} = \mathcal{H}_{\text{CE}}(\hat{y}, y), \tag{10}$$

where $\mathcal{H}_{\text{CE}}$ denotes the cross-entropy loss and $y \in \{0,1\}$ is the ground-truth binding label.

## 4 RESULTS AND DISCUSSION

In this section, we first describe the datasets used for training and evaluation, including an objective evaluation of explainability using the TCR-XAI benchmark. The pre-trained protein language models (PLMs) employed in our experiments are also presented, along with the procedures used to extract features from them. We then present and discuss the results of evaluation using standard metrics (i.e., ROC-AUC, accuracy) as well a metric (BRHR) designed to assess explainability. We conduct our experiments on carefully designed test sets that are meant to mimic TCR binding prediction on novel epitopes.

### 4.1 DATASETS AND BENCHMARKS

**Training Dataset and Test Dataset with Unseen Epitopes** To train and evaluate our model layers, we constructed a TCR-pMHC dataset comprising 349,716 paired samples of TCR alpha and beta chains, 2,316 unique peptides, 29,581 distinct CDR3a sequences, and 32,578 distinct CDR3b sequences. The dataset spans multiple species, primarily *Homo sapiens* and *Mus musculus*. Among the samples, 95.7% correspond to MHC-I and 4.3% to MHC-II. The dataset was compiled from VDJdb (Bagaev et al., 2020), TCR-McPAS (Tickotsky et al., 2017), IEDB (Vita et al., 2019), TBAdb Zhang et al. (2020), and 10x Genomics (10x Genomics, 2022). For all sources, we retained only samples that provided CDR3a, CDR3b, and peptide sequences. Any non-standard characters, irregular notations, or missing residues were discarded in the amino acid sequences. Negative samples were generated by directly sampling negative pairs for the 10x Genomics dataset, and for other datasets by randomly shuffling TCR and pMHC pairs. For each epitope, negative samples were generated at a ratio of 4:1 relative to positive samples. Finally, the dataset was split into training and test sets using a 95:5 ratio. The test set contains 15,503 samples spanning 288 epitopes that do not appear in the training data. To construct the evaluation set, we computed the Levenshtein distance between each pair of peptides. We then sampled a comparable number of peptides whose minimal pairwise distance exceeded different thresholds from 1 to 9, ensuring that all selected epitopes were excluded from and distinct from those in the training dataset.

**TCR-XAI Benchmark** Li et al. (2025b) introduced a benchmark to quantitatively assess explanation quality using residue-level contacts between TCR and pMHC, derived from 274 structural samples. For each sample, residue-level distances were computed in two ways: (1) from each CDR residue to the nearest atom in the peptide, and (2) from each peptide residue to the nearest atom in any CDR region. Smaller distances indicate stronger interactions and are treated as ground-truth for evaluating explanation methods.

## 4.2 PROTEIN LANGUAGE MODELS AND BASELINES

We extracted features from four pre-trained protein language models: ESM-1b (Rives et al., 2021), ESM-2 (Lin et al., 2023), and ProteinBERT (Brandes et al., 2022). ESM-1b is a 100M parameter model. ESM-2 provides variants ranging from 8M to 15B parameters (8M, 35M, 150M, 650M, 3B, 15B); due to resource limitations, we excluded the 15B variant. ProteinBERT provides a single 16M parameter model.

We consider two categories of comparable models in our evaluations. First, to evaluate PLM-based models, we constructed a standard linear classifier for PLM features. Specifically, we added two fully connected layers with hidden dimension three times the feature dimension and ReLU activation. Each classifier takes concatenated global representations of CDR3a, CDR3b, and the peptide as input and outputs a prediction score. For ProteinBERT, we used the provided global features, and for the ESM models, we averaged local residue-level features to obtain global representations. Second, we compared our models with two recent transformer-based TCR-pMHC prediction methods. MixTCRpred (Croce et al., 2024), one of the most widely used TCR-pMHC models, utilizes all CDR regions as input. We evaluated MixTCRpred on only CDR3 regions. TULIP (Meynard-Piganeau et al., 2024) is another recent model that outperforms the widely used NetTCR-2.2 (Jensen & Nielsen, 2023) baseline in terms of accuracy and generalization ability.

## 4.3 EXPERIMENTAL SETUP

All experiments were conducted on a Ubuntu server equipped with two NVIDIA A2000 GPUs, two Intel Xeon E5 CPUs, and 64 GB RAM. To enable efficient training and evaluation with large protein language models, we first extracted de-duplicated amino acid sequences from CDR3a, CDR3b and peptides. Features were then pre-computed using PLMs and stored. During model training and evaluation, amino acid sequences were indexed to retrieve and reassemble the corresponding features, allowing large-scale experiments to be performed within the memory constraints of two A2000 GPUs. Each model was trained for 150 epochs with batch size of 512 and a learning rate of $1 \times 10^{-3}$ using the AdamW optimizer. The dropout rate is 0.2 to ensure the generalization ability.

## 4.4 ROC-AUC ANALYSIS

| ROC-AUC | ProteinBERT | | ESM-1b | | ESM2-8M | | ESM2-35M | |
| --- | --- | --- | --- | --- | --- | --- | --- | --- |
| Top-$k$ | Linear | Ours | Linear | Ours | Linear | Ours | Linear | Ours |
| 100 | 0.772 | **0.999** | 0.900 | **0.982** | 0.830 | **0.926** | 0.783 | **0.960** |
| 150 | 0.675 | **0.895** | 0.795 | **0.854** | 0.719 | **0.786** | 0.685 | **0.822** |
| 200 | 0.625 | **0.792** | 0.716 | **0.759** | 0.658 | **0.708** | 0.632 | **0.735** |
| ROC-AUC | ESM2-150M | | ESM2-650M | | ESM2-3B | | MixTCRpred | TULIP |
| Top-$k$ | Linear | Ours | Linear | Ours | Linear | Ours | | |
| 100 | 0.713 | **0.985** | 0.876 | **0.960** | 0.846 | **0.969** | 0.906 | 0.821 |
| 150 | 0.633 | **0.860** | 0.762 | **0.836** | 0.732 | **0.841** | 0.773 | 0.706 |
| 200 | 0.593 | **0.765** | 0.690 | **0.746** | 0.668 | **0.749** | 0.698 | 0.648 |

Table 1: ROC-AUC comparison at Top-100, Top-150, and Top-200 peptides. Columns report results for PLM backbones (ESM-1b (Rives et al., 2021), ESM-2 (Lin et al., 2023), and ProteinBERT (Brandes et al., 2022)) with either a linear classifier or our method, with MixTCRpred (Croce et al., 2024) and TULIP (Meynard-Piganeau et al., 2024) included as reference baselines. Our methods significantly outperformed all other methods with all PLM backbones.

To assess the performance and generalization ability of our method, we report the ROC-AUC with the maximum false positive rate restricted to 0.1, which is a standard way for TCR-pMHC binding

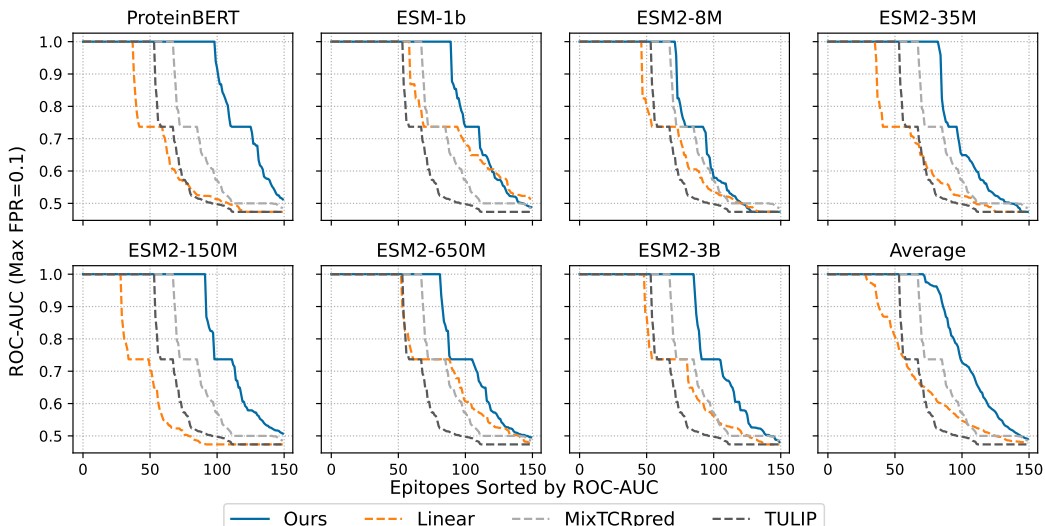

Figure 3: ROC-AUC with maximum false positive rate of 0.1 on the top-150 peptides in the test set. Results are reported for all PLM backbones (ESM-1b (Rives et al., 2021), ESM-2 (Lin et al., 2023), and ProteinBERT (Brandes et al., 2022)) with either a linear classifier or our method, and compared against MixTCRpred (Croce et al., 2024) and TULIP (Meynard-Piganeau et al., 2024) as comparable models.

prediction evaluation (Nielsen et al., 2024). The evaluation is conducted on the compiled test set with only peptides not observed during training. For each backbone model, we summarize the top-$k$ peptides with the highest ROC-AUC in Table 1, and present the ROC-AUC scores for the top 150 peptides in Figure 3.

As shown in Table 1 and Figure 3, all PLM backbones combined with our method outperform TULIP and MixTCRpred. In particular, ProteinBERT with our method achieves an ROC-AUC of 99.9% on the Top-100 epitopes, representing improvements of approximately 9% and 17% over MixTCR-pred and TULIP, respectively. Compared to linear classifiers, our method improves performance by about 20% on average with ProteinBERT and yields 8-20% gains with most ESM-2 backbones. With ESM-1b, our method performs 4-8% higher ROC-AUC than linear classifiers. We further evaluated our methods on the IMMREP23 benchmark (Nielsen et al., 2024), where they achieved ROC-AUC scores between 0.60 and 0.65. Overall, these results demonstrate that our method achieves competitive predictive accuracy and strong generalization, outperforming existing models across all backbones.

### 4.5 EVALUATION OF THE TCR-XAI BENCHMARK

| $a \to b$ | ProteinBERT | ESM-1b | ESM2 | | | | |
|---|---|---|---|---|---|---|---|
| | | | 8M | 35M | 150M | 650M | 3B |
| Peptide→CDR3a | 0.839 | 0.842 | 0.851 | 0.846 | 0.897 | 0.890 | 0.852 |
| Peptide→CDR3b | 0.842 | 0.877 | 0.812 | 0.858 | 0.850 | 0.823 | 0.885 |
| CDR3a→Peptide | 0.736 | 0.812 | 0.719 | 0.712 | 0.773 | 0.815 | 0.742 |
| CDR3b→Peptide | 0.790 | 0.813 | 0.782 | 0.766 | 0.792 | 0.806 | 0.815 |

Table 2: Binding Region Hit Rate ($t$=0.25) across different PLM backbones for Peptide→CDR3a, Peptide→CDR3b, CDR3a→Peptide, and CDR3b→Peptide, where $a \to b$ denotes $a$ contacts to $b$.

We evaluate the explanation quality of the contact prototypes using the TCR-XAI benchmark (Li et al., 2025b). The Binding Region Hit Rate (BRHR) (Li et al., 2025b) quantifies the proportion of true binding residues, defined by structural proximity, that are correctly identified by an explanation method. More concretely, for a chosen percentile threshold $t$, we compare the top $t$ fraction of residues ranked by contact scores and the top $t$ fraction of residues ranked by pairwise distance (between peptide and chosen TCR chain). A residue is counted as a *hit* if is in both sets (i.e., it is

considered important for prediction and has close structural proximity). For our experiments, we use $t = 0.25$ because it is the most restrict threshold but ensuring each case has at least one contact residue. The hit rate is computed per sequence type for each positive sample, and the final BRHR is reported as the average over the TCR-XAI benchmark. On TCR-XAI benchmark, the different PLMs with our TCR-EML can achieve 71.4% accuracy in average.

Table 2 presents the BRHR results across PLM backbones for Peptide→CDR3a, Peptide→CDR3b, CDR3a→Peptide, and CDR3b→Peptide interactions, where $a \rightarrow b$ denotes contacts from $a$ to $b$. For peptide to CDR3 interactions, all backbones with contact prototype layers achieve BRHR values above 0.71. For peptide to CDR3 interactions, BRHR values exceed 0.81 across all PLMs with contact prototypes. These results indicate that our contact prototype layers provide reliable residue-level explanations of CDR3-peptide interactions in TCR-pMHC binding prediction.

## 4.6 CASE STUDY

To illustrate the practical use of our model, we present a case study on a self-antigen associated with rheumatoid arthritis. Specifically, we analyze the HLA-DR4-bound citrullinated peptide vimentin-64cit59-71 (PDB: 8TRR) (Loh et al., 2024). Using ProteinBERT in combination with our TCR-EML, which demonstrates superior performance and generalization, we visualized contact distance weights between the peptide and TCR. To summarize peptide interactions, we computed an integrated contact scores by averaging the contact scores of CDR3a and CDR3b.

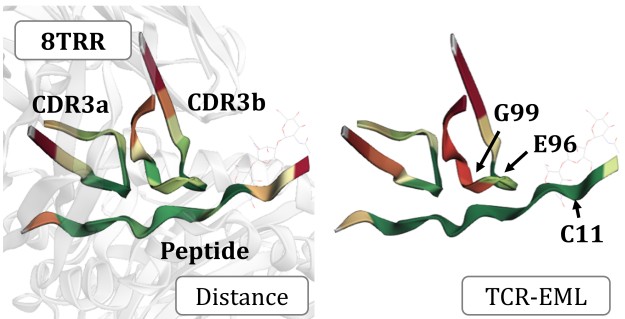

Figure 4: Predicted versus experimental peptide-CDR3 contact distances for HLA-DR4-bound vimentin-64cit59-71 (PDB: 8TRR) (Loh et al., 2024). TCR-EML predictions closely match experimental contacts, highlighting the model explainability.

As shown in Figure 4, the contact distances predicted by our method closely match the experimentally determined distances, with only minor deviations. The BRHR ($t = 0.25$) achieves 1.0 for peptide and CDR3a, and 0.67 for CDR3b. Notably, in CDR3b, TCR-EML correctly identifies the E96 contact region but misses G99, which leads to the lower BRHR for this region. On the peptide side, our method assigns a high weight to C11, which is not a true contact site. These minor discrepancies notwithstanding, the results indicate that TCR-EML provides high-quality, interpretable explanations that faithfully capture biologically relevant contact regions.

## 4.7 CONTACT PROTOTYPE ANALYSIS

.5To examine the explanations provided by the contact prototype layers and their relationship to actual binding among all samples, we collected contact prototype layers for all PLM backbones with our TCR-EML. Then, for each group of the contact scores, ensuring that the valid distance scores are aligned at the center of the padded contact scores. All distance maps are grouped by negative and positive prediction, and then averaged to reveal the contact patterns captured by the contact prototype layers.

As shown in Figure 5, high-contact regions are concentrated near the center of the contact scores, around the 8-mer position, which corresponds to the typical length of a peptide or CDR3 region. For positive samples, the aver-

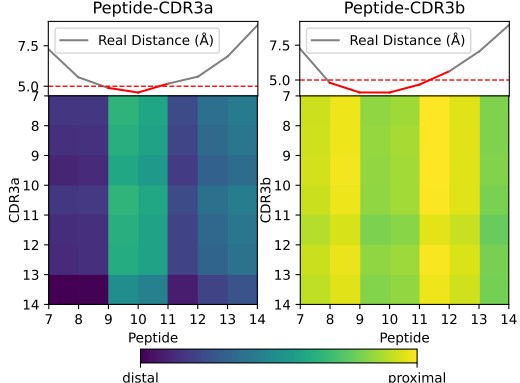

Figure 6: Average contact scores from Protein-BERT contact prototype layers. Zoomed view highlights distinct binding patterns.

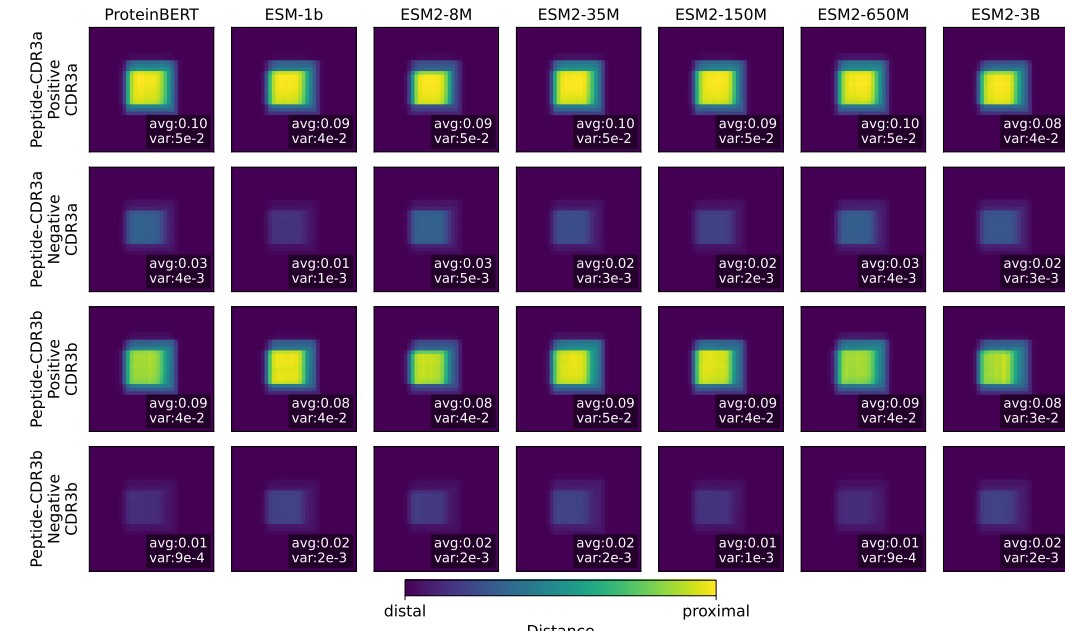

Figure 5: Average contact scores from ProteinBERT contact prototype layers. Positive samples show proximal contacts, whereas negative samples exhibit distal contacts.

age contact distance scores range from 0.08 to 0.10, whereas for negative samples, they fall between 0.01 and 0.03. This demonstrates that our model effectively distinguishes binders from non-binders through contact region patterns. Moreover, the variance further highlights this difference: positive samples exhibit variances greater than $10^{-3}$, while negative samples remain below this threshold. This suggests that the model assigns uniformly small values to non-binders, but identifies diverse and biologically meaningful contact residues for binders.

To investigate the contact prototype patterns in greater detail, we use ProteinBERT as a representative example, since it achieved the best performance and generalization in the unseen epitope evaluation. As shown in Figure 6, we present zoomed views spanning residues 7-14 for both CDR3 and peptide sequences, given that most peptides and CDR3 regions are shorter or equal than 8 residues. We also computed average distances from experimental structures for comparison. All distance maps were padded to a uniform length to ensure that valid regions were centered within the maps.

Notable differences emerge between peptide-CDR3a and peptide-CDR3b interactions. According to the experimental structures, peptide-CDR3b exhibits a broader and closer contact area, which is consistent with our method assigning higher contact scores to CDR3b. For CDR3a, residues with average experimental distances below 5 Åare primarily located at residues 9 and 10; our method likewise assigns these positions the highest scores. For CDR3b, the main experimental contact regions are residues 8-11. Our model highlights residues 8 and 11 while also assigning high scores to residues 9 and 10. Overall, these results demonstrate that our method successfully recapitulates contact distance patterns observed in experimental structures.

## 5 CONCLUSION

In summary, we present TCR-EML, explainable model layers for TCR-pMHC binding prediction, designed to improve both explainability and generalization to unseen epitopes. TCR-EML can be used with pre-trained protein language model (PLM) backbones without additional fine-tuning or retraining, thereby transforming them into explainable TCR-pMHC binding predictors. Across experiments, PLMs equipped with TCR-EML outperform both linear classifiers and state-of-the-art baselines, including MixTCRpred and TULIP. Through a case study on an MHC-II TCR–pMHC complex, analysis of contact prototype patterns, and evaluation on the TCR-XAI benchmark, we show that TCR-EML provides accurate and biologically meaningful explanations, validated against experimental structures.

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

## A  REPRODUCIBILITY STATEMENT

To comply with the double-blind review policy while ensuring reproducibility, we provide a self-contained code package with detailed training instructions. Model weights are not included due to their large size; however, the protein language model weights can be obtained from publicly available repositories referenced in our documentation. After publication, we will release a public repository containing the full code and trained weights.

## B  LARGE LANGUAGE MODEL USAGE STATEMENT

We employed large language models (LLMs), primarily ChatGPT, in two limited ways:

- as a coding assistant, and
- for polishing written text.

**Coding Assistant**  LLMs were consulted to clarify documentation, organize API references, and suggest debugging strategies. All code, documentation, and fixes obtained were manually reviewed and verified by the authors.

**Polishing Article**  LLMs were used only to refine the clarity and style of sentences written by the authors and to format tables from raw data. No raw text or substantive content was generated by LLMs. All refined content was manually checked and further revised by the authors.

