# OpenReview forum: "TCR-EML: Explainable Model Layers for TCR-pMHC Prediction"
_ICLR.cc/2026/Conference — ICLR 2026 Conference Withdrawn Submission_

### Official Review · Reviewer_DJwy · 2025-11-01

**Soundness:** 3
**Presentation:** 2
**Contribution:** 2
**Rating:** 2
**Confidence:** 3

**Summary:**

This paper presents TCR-EML, an "explain-by-design" module for TCR-pMHC binding prediction, which addresses the problem of standard transformer models being black boxes. The proposed layers are designed to be incorporated into existing, pre-trained protein language model backbones, enabling interpretability without requiring complete retraining. TCR-EML consists of a cross-attention-based Feature Enhancement and Fusion block and "Contact Prototype" layers that explicitly model amino acid residue contacts. This architecture directly links the model's prediction to known binding mechanisms, and it demonstrates competitive predictive accuracy on large-scale datasets while achieving high explainability scores on the quantitative TCR-XAI benchmark.

**Strengths:**

- By using "Contact Prototype" layers , the model's output is an explanation, rather than relying on post-hoc methods which can be unfaithful to the model's internal logic. The final prediction is a direct function of the predicted contact area scores.
- The approach is designed as a lightweight prediction head that can be added to general-purpose PLMs like ESM-2, without needing retraining.
- The model shows good results, outperforming existing baselines like MixTCRpred and TULIP in ROC-AUC analysis, especially on unseen epitopes.

**Weaknesses:**

- The method is described as a general approach to addressing interpretability of transformers, but is only evaluated on a niche task of TCR-pMHC binding prediction.
- For most of the training data, negative samples were generated by "randomly shuffling TCR and pMHC pairs". This method creates simple negatives that are not biologically realistic, which can artificially inflate predictive performance metrics.
- While the explainability provided is an interesting proposition, the paper's case study shows imperfections. The authors note that TCR-EML correctly identifies one contact region but misses others or incorrectly assigns high weights to some residues on the peptide.

**Questions:**

- Did the authors consider evaluating their model on other protein language modeling tasks?
- The large training dataset is composed mostly of MHC-I data, with only a small fraction of samples corresponding to MHC-II. Did the author perform some class rebalancing during training to address this?

---

### Official Review · Reviewer_weB2 · 2025-11-01

**Soundness:** 3
**Presentation:** 2
**Contribution:** 3
**Rating:** 8
**Confidence:** 3

**Summary:**

The paper proposes adding “explainable model layers” downstream pretrained protein language models in order to enhance biological interpretability of the features extracted on the task of TCR-pMHC binding prediction, eventually improving performance. The data is made of triplets of CDR3a, CDR3b and peptide sequences. The explainable layers are a Feature Enhancement and Fusion block, which is supposed to learn cross correlations between alpha and beta chains and the peptide sequence through cross-attention, and contact prototype layers, which identify residues in contact by measuring similarity of their fused embeddings. The method is tested on an annotated dataset with various pretrained upstream models, showing increase in performance and interpretability.

**Strengths:**

The method works without the need of retraining the upstream protein language model, making it flexible. It includes information from both alpha and beta chains. It provides robust increase in performance w.r.t. state of the art methods for TCR-epitope binding predictions (Table 1, Fig. 3). The way contacts are inferred is reliable (Table 2, Fig. 4-6).

**Weaknesses:**

The way the FEF block works and the kind of correlations it captures are not very clearly explained. For other questions, see below.

**Questions:**

- Can the authors explain more in detail the principles behind fused embeddings?
- The authors train the model with a 4:1 imbalance ratio between negative a positive pairs, and then class-reweight the loss to account for this ratio. Is there a specific reason for this design choice, given the amount of negative data to include in the training set can be decided?
- How do different negative designs (true negative pairs vs shuffled pairs) impact predictions? See for example [1]
- Is the method flexible if some of the data modalities (e.g., the alpha chain) are missing?

[1] Ursu et al. Training data composition determines machine learning generalization and biological rule discovery, Nat. Mach. Int. 7, 1206–1219 (2025)

Minor:

Line 417, “.5To” → “To”

---

### Official Review · Reviewer_E7iy · 2025-11-01

**Soundness:** 1
**Presentation:** 1
**Contribution:** 1
**Rating:** 0
**Confidence:** 4

**Summary:**

The paper proposes a head for better understanding the reasons why the PLM will do the TCR-pMHC relevant prediction.
The head design is based on the prototype learning, and the experimental results also suggest the effectiveness of the proposed method.

**Strengths:**

- The motivation to make the PLM explainable is meaningful. And exploring the T-cell mechanism is also important for life science.

**Weaknesses:**

- The writing and presentation are inferior.
- MHC appearing in the abstract should be written with the full name,
- A lot of terms that have nothing to do with learning representation appear in the introduction without explanation (e.g., MHCI, MHCII, CD8+ T cells, CD4+ T, PLM). This makes the paper hard to understand.
- A lot of important terms appear for the first time, but are not explained (e.g. CDRα and CDRβ, CDR3a, CDR3b).
- Appending a prototype layer to make the PLM explainable seems overclaim. The only explanation result is the contact score (distance), which is weak and cannot provide important insight. TSNE for the PLM embedding may get similar results. Therefore, the contribution of this paper is limited.

**Questions:**

- For the second paragraph of the introduction, it would be better to provide a figure to illustrate the research objects/concepts and the research question.
- The presentation and writing need to be improved.

---

### Official Review · Reviewer_uG19 · 2025-11-01

**Soundness:** 3
**Presentation:** 3
**Contribution:** 2
**Rating:** 4
**Confidence:** 3

**Summary:**

Interpretability of large, black-box machine learning models is an important and timely issue. The authors plug "explain-by-design" layers on top of protein sequence representations extracted by pretrained language model embeddings such as ESM. The paper is well-written and computational experiments are clearly presented, but the actual gain in explainability could be justified more.

**Strengths:**

- The paper motivates interpretability in TCR–pMHC prediction well.
- Architecture (“TCR-EML”) is straightforward: plug explainable prototype layers on top of pretrained protein language model embeddings.
- Demonstrates strong empirical results on public datasets and the TCR-XAI benchmark.
- Quantitative metrics (ROC-AUC, BRHR) are presented clearly; figures are decent.
- The paper includes reproducibility and LLM-usage statements, as well as source code.

**Weaknesses:**

- The proposed “explainable model layers” are a minor architectural variant, essentially combining cross-attention and similarity-based prototype aggregation. There is little theoretical or methodological depth beyond known ideas in prototype learning and cross-attention.
- Comparisons are limited: only MixTCRpred and TULIP are used as baselines, both relatively new and not necessarily the strongest available. No ablation on the design choices (e.g., thresholds, temperature).
- The use of PLMs is plug-and-play; there is no fine-tuning or adaptation beyond feature extraction, so it’s unclear how much of the performance comes from the PLMs themselves and how much from the plugged explainable layer.
- Explainability evaluation via BRHR (Binding Region Hit Rate) is shallow: the metric correlates with contact proximity but does not assess causal or mechanistic interpretability.
- Besides contact prediction, the authors don't present additional justification for actual "explainability". Contact prediction is already seen in the self-attention matrices of protein LLMs.
- Some results are implausibly high (ROC-AUC = 0.999 for ProteinBERT, Table 1 p. 6), suggesting possible data leakage or inadequate negative sampling. Please check.

**Questions:**

- In page 8, line 418, what is ".5To" ?
- How was the “case study” (Figure 4 p. 8) selected? Is it a random choice or was it cherry-picked?
- Besides contact prediction, the authors could present additional justification for "explainability". Are there other explainable factors aiding the predictions?

---

### Official Review · Reviewer_nxmS · 2025-11-11

**Soundness:** 2
**Presentation:** 2
**Contribution:** 2
**Rating:** 4
**Confidence:** 5

**Summary:**

The authors proposed TCR-EML, with two explainable model layers that can be integrated into protein language model architectures for TCR–pMHC binding prediction. The method incorporates prototype layers representing amino acid contact patterns derived from biochemical binding mechanisms. Experiments on large-scale datasets demonstrate improved predictive performance andmodel explainability compared with prior methods.

**Strengths:**

Introduces TCR-EML, explainable model layers for TCR–pMHC binding prediction with Plug-and-play compatibility with pre-trained protein language model (PLM) backbones, requiring no additional fine-tuning or retraining.

Provides biologically meaningful explanations through analysis of contact prototype patterns.

Validated with a case study on an MHC-II TCR–pMHC complex and TCR-XAI benchmark evaluation, showing alignment with experimental structural data.

**Weaknesses:**

(1) The proposed method introduces two additional components — a sequence feature layer and a contact strength calibration layer — to the base TCR–pMHC prediction model. However, this design appears somewhat ad hoc and shows limited conceptual novelty. The first layer resembles the EGM approach, while the second shares similarities with PISTE (the M*A part, in particular).

(2) In the evaluation, I think there should more baselines included from the literature of TCR–pMHC binding prediction;

(3) For TCR–pMHC binding prediction with imbalanced data, AUROC can be misleading because the majority non-binding class dominates. Better metrics focus on the minority positive class and include precision, recall, and F1-score, and top-k accuracy; area under the precision-recall curve (AUPRC) is also informative for rare binders; and Matthews correlation coefficient (MCC) or balanced accuracy, which account for class imbalance.

**Questions:**

(1) This paper appears to focus primarily on enhancing the performance of protein language models (PLMs) for TCR–pMHC binding prediction. Regarding interpretability, it would be helpful to clarify what additional insights the proposed method provides beyond existing approaches that visualize contact probability maps. Specifically, can the model yield biologically meaningful interpretations or mechanistic insights that are valuable for advancing immunological research?

(2) I am wondering why you do not need to re-train the backbone model, since the two new layers added will revise the loss function.

(3) The authors state that current algorithms for TCR–pMHC binding prediction are typically black-box transformer models that lack interpretability and do not provide clear rationales for their predictions, and that that PISTE provided limied explanability,  and requires additional steps to extract explanations. This statement is not entirely accurate. For example, the PISTE model (Feng et al.) can accurately predict both binding status and residue-level contact maps without relying on any 3D structural data during training.

(4) It's worthwhile to note that, if you used embedding from PLMs in the context of two or more sequence interesting with each other, their positional encoding is in fact physically not quite meaningful because the relative positions in such a interesting system is not known and cannot simply be encoded by existing positional encoding schemes like sine/cosine (adding positions with embeddings leads to noice attention scores) or ROPE, nor even learnable positions (which do not behave like real positions in 3D).

---

### Note · Authors · 2025-11-21

I have read and agree with the venue's withdrawal policy on behalf of myself and my co-authors.